# Advances Towards Ex Situ Conservation of Critically Endangered *Rhodomyrtus psidioides* (Myrtaceae)

**DOI:** 10.3390/plants14050699

**Published:** 2025-02-24

**Authors:** Lyndle K. Hardstaff, Bryn Funnekotter, Karen D. Sommerville, Catherine A. Offord, Ricardo L. Mancera

**Affiliations:** 1Curtin Medical School, Curtin University, GPO Box U1987, Perth, WA 6845, Australia; 2The Australian PlantBank, Botanic Gardens of Sydney, Mount Annan, NSW 2567, Australia; karen.sommerville@botanicgardens.nsw.gov.au (K.D.S.); cathy.offord@botanicgardens.nsw.gov.au (C.A.O.); 3Kings Park Science, Department of Biodiversity, Conservation and Attractions, Kings Park, WA 6005, Australia; bryn.funnekotter@dbca.wa.gov.au

**Keywords:** exceptional species, in vitro culture, plant tissue culture, cryopreservation, cryobiotechnology, myrtle rust

## Abstract

*Rhodomyrtus psidioides* (G.Don) Benth. (Myrtaceae) is a critically endangered rainforest species from the east coast of Australia, where populations have severely and rapidly declined due to the effects of repeated myrtle rust infection. With very limited material available in the wild and freezing-sensitive seeds that have prevented storage in a seed bank, ex situ conservation of this exceptional species has proven difficult. Material from a seed orchard grown at the Australian Botanic Garden Mount Annan was successfully used to initiate three new accessions into tissue culture from cuttings, and to undertake cryopreservation experiments using a droplet-vitrification (DV) protocol for both seeds and cultured shoot tips. Use of seedling material for tissue culture initiation was very effective, with a 94–100% success rate for semi-hardwood explants and a 50–62% success rate for softwood explants. Although no survival of seeds after cryopreservation was observed, seeds of *R. psidioides* showed some tolerance of desiccation and exposure to cryoprotective agents. Regeneration after cryopreservation using a DV protocol was demonstrated in only one shoot tip precultured on basal medium containing 0.4 M sucrose and incubated in PVS2 for 20 min prior to immersion in liquid nitrogen. These results demonstrate the value of living collections in botanic gardens for conservation research, highlight the importance of germplasm choice for tissue culture initiation, and demonstrate the potential of cryobiotechnologies for the ex situ conservation of exceptional plant species.

## 1. Introduction

*Rhodomyrtus psidioides* (G.Don) Benth. (family: Myrtaceae), the native guava, is a shrub or small tree found in sub-tropical rainforest, littoral rainforest, and rainforest margins in eastern Australia [1]. Not considered threatened before the arrival of the myrtle rust-causing pathogen *Austropuccinia psidii* in 2010 [2], *R. psidioides* has since demonstrated extreme susceptibility to infection [3] and was listed as critically endangered in 2020 as a direct result of drastic population decline caused by myrtle rust [4]. The species is now considered an emergency priority for ex situ conservation [5].

Seed banking was initially considered as a conservation option for this species, but the oily seeds (Figure 1) were found to have intermediate storage behaviour, being tolerant of drying but sensitive to freezing [1]. In addition, viable fruit and seeds are no longer being produced in the wild due to the effects of myrtle rust [6]. These factors classify *R. psidioides* as an exceptional species under two categories—seed not able to be collected (Exceptionality Factor 1) and seed not able to be stored (Exceptionality Factor 3) [7,8]. The development of alternative conservation strategies is, therefore, essential.

The susceptibility of *R. psidioides* to myrtle rust was first documented within months of the arrival of *A. psidii* in Australia [9] and tree death due to infection was reported not long after [10]. In response to these reports, early conservation research from 2015 focused on the development of vegetative propagation techniques, tissue culture of wild and cultivated specimens at the Botanic Gardens of Sydney, and assessment of seed viability and suitability for seed banking [2]. A seed orchard established at the time has been essential to the production of seed for research in the absence of wild-sourced material [11]. The continued decline of *R. psidioides* in the face of myrtle rust, the growing cost of maintaining living collections, and the difficulty of banking seeds using conventional techniques highlight the need for ex situ conservation strategies such as tissue culture and cryopreservation to ensure the persistence of the species.

The use of cryobiotechnologies for the conservation of Australian Myrtaceae species has been limited, particularly for species without commercial value [12]; however, the initial success of *R. psidioides* in tissue culture was an encouraging sign for further conservation efforts [2]. The purpose of the present research was to advance ex situ conservation knowledge for *R. psidioides* by: (1) investigating whether it is possible to preserve the freezing-sensitive seeds in liquid nitrogen using a cryoprotection protocol; (2) investigating the relative benefit of softwood versus semi-hardwood cuttings for initiating cultures of new accessions; and (3) developing cryopreservation protocols for long-term conservation of tissue-cultured shoot tips.

## 2. Results

### 2.1. Seed Cryopreservation

In tests comparing the tolerance of seeds to different sterilant solutions and basal media, bacterial contamination emerged on 80–100% of seeds within five days after sowing on all basal media, regardless of the sterilant solution used. The experiment was terminated when none of the seeds had germinated after 76 days, at which point seed fill was found to be low for all treatment combinations (17–67%). Most of the filled seeds were assessed as ‘mushy’ and were unlikely to have germinated with extended monitoring. As a result of this experiment, the seed collection was reprocessed to ensure only filled seeds were used for the cryopreservation experiment.

Contamination continued to be an issue when cryopreserving seeds, with low levels of fungal contamination emerging even during the 48 h preculture period. Four days after cryopreservation, contamination had emerged on all *R. psidioides* seeds of every treatment combination, both those immersed in LN and those that were not. Contamination covered most of the seeds by day 18, and after one month, all Petri dishes were completely covered by a thick mat of microbial contamination.

There was a significant difference in survival after two months between *R. psidioides* seeds subjected to different treatment combinations (ANOVA F_(11,48)_ = 2.172, *p* = 0.032; Figure 2), which a generalized linear model (GLM) indicated was driven by whether or not the seeds were immersed in LN (*df* = 1.55, χ^2^ = 11.347, *p* = 0.001) as no cryopreserved seeds regenerated (Table 1). The highest rate of survival was noted for seeds that were precultured on the lowest desiccation treatment (0.09 M sucrose basal medium) and not treated with PVS2, with a mean survival rate of 32% (Figure 2). There was no significant difference in regeneration after two months between *R. psidioides* seeds subjected to different treatment combinations (ANOVA F_(11,48)_ = 1.983, *p* = 0.052; Table 1). Regeneration rates ranged from 16 to 24% among successful treatments (Table 1), and while treatment with PVS2 (30 min PVS2 vs. 0 min PVS2) did not appear to further reduce regeneration after desiccation (Figure 2), there was also no clear relationship between desiccation treatments and regeneration as all treatments resulted in significant severe declines in survival at day 60 (Figure 2). Of the germinated seeds, 59% survived removal from contaminated Petri dishes and planting into potting mix (Table 1). Surviving seedlings continued to grow well for seven months, demonstrating no adverse reaction to desiccation or cryoprotective treatments.

### 2.2. Tissue Culture Initiation

The initiation of *R. psidioides* explants into tissue culture was successful for all three accessions. Explants were monitored for one month after initiation, during which time no visible contamination was apparent. Higher survival rates were noted for explants derived from semi-hardwood cuttings with axillary buds (94–100%; Table 2), compared with explants derived from softwood cuttings with either apical or axillary buds (50–62%). Collections continued to grow and multiply well (Figure 3), with leaf drop only noted for some explants. A trial two years after initiation added 2 g/L activated charcoal to the subculture medium, which resulted in good shoot growth, colour, and increased root production compared to regular medium. However, explant health appeared to reduce during the consecutive subculture with charcoal and the decision was made to use this medium only once every six months.

### 2.3. Shoot Tip Cryopresevation

Three weeks after cryopreservation, there was a significant difference in the survival of *R. psidioides* STs among different treatment combinations (ANOVA F_(24,24)_ = 13.62, *p* < 0.001; Figure 4). A GLM indicated that this was driven both by desiccation treatment (*df* = 3.45, χ^2^ = 33.516, *p* = 0.020) and whether the STs were immersed in LN (*df* = 1.41, χ^2^ = 16.993, *p* < 0.001). There was also a significant difference in regeneration (production of new leaves) among *R. psidioides* STs subjected to different treatment combinations (ANOVA F_(24,24)_ = 24.93, *p* < 0.001). Again, a GLM indicated this was driven both by desiccation treatment (*df* = 3.45, χ^2^ = 32.069, *p* = 0.008) and whether the STs were immersed in LN (*df* = 1.41, χ^2^ = 6.350, P < 0.001). STs that were precultured on 0.8 M sucrose basal medium and those that were immersed in LN had lower rates of both survival and regeneration. Shoot tip survival (presence of green tissue) after immersion in LN was seen in multiple treatment combinations (Figure 4), including preculture on WA and incubation in PVS2 for 20 and 60 min, preculture on 0.4 M sucrose and incubation in PVS2 for 20 min, and preculture on 0.8 M sucrose and incubation in PVS2 for 30 min. However, the only ST (17%, *n* = 6) to regenerate after immersion in LN was precultured on 0.4 M sucrose basal medium and incubated in PVS2 for 20 min, although the growth of this ST (Figure 5b) was poorer than that of a control ST (Figure 5a). No regenerated ST developed any further or grew more leaves than those pictured in Figure 5.

## 3. Discussion

Ex situ conservation strategies are essential for maintenance of biodiversity given that 45% of flowering plant species are likely in danger of extinction globally [13]. Seed banks have proven invaluable for the ex situ conservation of many agricultural and wild species; however, the development of alternative conservation approaches for exceptional species is essential given the increasing awareness of the characteristics that prevent their long-term storage in seed banks [14,15]. Exceptional species include those which have seeds that are intolerant of the desiccation or freezing required for storage in a conventional seedbank, as well as those species no longer producing seed in the wild [7]. Cryobiotechnologies such as tissue culture and cryopreservation are the only options to effectively store a range of plant germplasm types and genetic diversity of exceptional species in the long-term [15]. The need for the development of effective conservation protocols is particularly urgent when external threats to ex situ living collections cannot be controlled, as is the case for *R. psidioides* and other threatened species affected by airborne pathogens such as myrtle rust. Here, we report several advances in tissue culture and cryopreservation knowledge for *R. psidioides* that will provide a firm foundation for long-term conservation of the species.

The sterilization of fresh seeds proved to be very difficult, with heavy loads of fungal or bacterial contamination developing rapidly after sterilization regardless of sterilant concentration or the presence/absence of sucrose in the germination medium. The difficulty in producing clean seeds is likely related to the warty texture of the seed coat (Figure 1c) and the presence of a sticky resinous substance on the seed surface. Longer sterilization periods, stronger sterilant concentrations, or the application of fungicides and bactericides may be needed to resolve the problem. Alternatively, sowing seeds directly into seed raising mix may be a better recovery option for species like *R. psidioides*, with greater airflow, reduced humidity, and reduced sugars and nutrients likely to limit fungal and bacterial growth compared to recovery under in vitro conditions.

Although seed collections for this study were made from cultivated specimens, the collection of sufficient healthy seed for experimentation was a limiting factor for the number of treatments able to be applied. *Rhodomyrtus psidioides* is known to have a variable and often low seed set [2,16], and the very poor seed fill of preliminary collections made in 2019 was likely caused in part by the severe drought across eastern Australia in the 24 months prior to seed collection [17]. Better seed collections were possible in 2020 due to good management of cultivated specimens and were vital for this study given that this critically endangered species is no longer producing viable fruit and seeds in the wild [6].

While seeds affected by contamination did germinate in vitro and often continued to grow through a thick mat of contamination, they subsequently died if not transferred to potting mix within a few days of germination. In those cases, fungi or bacteria may have predated the seed directly, or contamination may have altered the basal medium through consumption of certain elements, the addition of leachates, or by changing the pH and causing it to become inhospitable for germinants [18]. Although seed material is preferred for the ex situ conservation of maximum genetic diversity, these issues with contamination may be avoided by using ST material from established in vitro cultures for cryopreservation [19].

The very successful initiation of tissue cultures from three new accessions demonstrated the benefit of establishing a living collection that can then be used to generate seeds and juvenile plant material for culture. The establishment of such a collection is likely to be key to long-term conservation for other Myrtaceous species severely impacted by myrtle rust. The lack of contamination on either softwood or hardwood explants during initiation demonstrated that the original sterilization protocol [2] was very effective for juvenile material grown and maintained under nursery conditions, while the greater initiation success with semi-hardwood (94–100%) compared to softwood (50–62%) explants demonstrated the benefit of trialling initiation from different tissue types. The initiation rate for semi-hardwood explants from 3- to 6-month-old seedlings in this study (94–100%) greatly exceeded that of the previous study [2] in which semi-hardwood explants were taken from a 1-year-old seedling (56%), a mature garden specimen (40%) and 3-month-old rooted cuttings from a wild plant (11%). The greater success in this study may have been due to a reduced contaminant load, and perhaps more active cell division, in the younger plants.

Genetic diversity is an essential component of the value of ex situ collections for conservation, but high levels of diversity can be a difficult goal to achieve [20]; hence, most ex situ collections of wild species contain extremely limited diversity in comparison to those of agricultural species [14]. An aim of this work was to increase the genetic diversity of *R. psidioides* in tissue culture with a view to future long-term conservation by cryopreservation. The three new accessions initiated in 2020 have continued to multiply well over a number of years, adding to the diversity of material available for further research. However, improving the diversity in a living collection also increases the maintenance requirements, particularly for tissue culture collections that require labour-intensive and regular subculture, which may amplify the likelihood of future contamination [19,21,22,23,24]. Hence, this work also aimed to develop cryopreservation protocols for seed and shoot tip material which would reduce ongoing maintenance requirements and risk of fungal or bacterial infection in comparison to tissue culture collections [21,23,25,26].

Survival (a visual assessment of germplasm health) and recovery (the production of a root and/or shoot) after cryopreservation varied between germplasm types and among cryopreservation treatments. While no survival or regeneration was observed for seeds of *R. psidioides* immersed in LN after any treatment, there was 17% survival (*n* = 6) for fresh shoot tips (STs) incubated in PVS2 for 20 and 60 min and for STs desiccated on basal media containing 0.4 and 0.8 M sucrose and incubated in PVS2 for 20 and 30 min, respectively. A single ST desiccated on basal medium containing 0.4 M sucrose and incubated in PVS2 for 20 min was the only one to regenerate (17%, *n* = 6) after immersion in LN. These positive findings will be the building blocks for future cryopreservation experiments. It is known that cryopreservation success is dependent on the type of germplasm used, for example, *Eucalyptus* callus appears to survive immersion in LN better than buds [27], and *R. psidioides* STs performed better than seeds in this study. Optimized protocols are also likely to be required for each cultivar or genotype of the same species [28]. For example, Kaya et al. [29] found higher regeneration of *Eucalyptus* cultivars using a DV protocol compared with an encapsulation-dehydration protocol; however, their success varied among cultivars.

While there was little to no regeneration of seeds or shoot tips after immersion in LN, high survival rates (up to 32%) for many of the treatments that did not include immersion in LN suggest that *R. psidioides* has some tolerance of desiccation and cryoprotective agents, and that cryopreservation may be a viable option for the conservation of this exceptional species if pretreatment and recovery steps can be optimized to increase regeneration and ongoing survival. Further protocol modifications are needed to improve germplasm survival and regeneration after cryopreservation, which may include: increased incubation times in the plant vitrification solutions in order to further reduce freezable water content without the damage associated with desiccation alone [30,31], addition of antioxidants in pre-treatment and preculture media to alleviate oxidative stress [32], reduced concentration of cryoprotective agents using alternative vitrification solutions [33], or a two-step preculture with graduated increases in sucrose concentration [33,34].

Other methods to improve ST survival and regeneration may be taken from the horticultural practice of hardening plants. For example, cold hardening has been used to increase frost tolerance of seedlings of *Eucalyptus* species [35], and cold hardening of in vitro cultures has been used to increase post-cryopreservation recovery in apples [36] and potatoes [37]. Cold hardening of in vitro cultures has been shown to improve post-cryopreservation recovery in Australian native species such as *Lomandra sonderi* [38] and *Grevillea scapigera* [39], and early results suggest that it may also be an effective way to increase survival of *Gossia fragrantissima* STs after immersion in LN [40]. The optimization of recovery conditions, particularly basal media, is also critical for germplasm regeneration after cryopreservation [41]. Thus, future cryopreservation research for *R. psidioides* and other endangered Myrtaceae species will focus on the optimization of recovery basal media for different germplasm types.

Although the majority of Myrtaceae are thought to have seed that can be stored in standard seed banking conditions [42], alternative conservation strategies such as tissue culture and cryopreservation are required for the high number of exceptional species within the family. The development of effective cryopreservation protocols, especially when material choice is informed by an understanding of genetic diversity in populations to maximize the value of ex situ collections [43], is likely to be instrumental in the conservation of exceptional Myrtaceae species such as *R. psidioides* given the ongoing impacts of myrtle rust and the imminent risk of extinction for many species [5] in this globally significant family. Although this study demonstrates no regeneration of seeds and limited regeneration of shoot tips after cryopreservation, these results indicate the potential of cryopreservation as a conservation solution for *R. psidioides*. This study indicates that cryopreservation of shoot tips is likely to be a better solution for conservation of *R. psidioides* than seeds, particularly given the effects of myrtle rust on seed production in the wild; however, further research is required to optimize protocols to improve regeneration after cryopreservation.

## 4. Materials and Methods

### 4.1. Processing and Sterilization of Seeds

Ripe fruits of *R. psidioides* were collected from a cultivated specimen at the Australian Botanic Garden Mount Annan (ABGMA) in March 2020. Seeds were extracted within one week of fruit collection and used for experimentation within two weeks. After removing the skin and persistent calyx, the flesh was gently pulped by hand and submerged in a beaker containing 5 mL/L of concentrated liquid pectolytic enzyme (Everzym Liquid, Ever #A003A07K0001, Treviso, Italy). The pulp was left to break down while stirred by a paint mixer for 3.5 h with intermittent hand agitation to assist in the process [44]. The mixture was then rinsed with fresh water until only the seeds remained, and the seeds were spread on paper towel and allowed to dry overnight in ambient laboratory conditions at 23 °C. Tolerance of *R. psidioides* seeds to four sterilant solutions and germination on five basal media (Table 3) were then compared. Seeds were sterilized for 5 min in 0%, 0.5%, 1% and 2% sodium hypochlorite (NaClO) solutions prepared with White King^®^ Premium Bleach (a household bleach containing 4% NaClO; Pental, Shepparton, Australia) and sterile deionized water.

### 4.2. Cryopreservation of Seeds

Whole seeds (1.5–2 mm long, Figure 1c) were cryopreserved using a droplet-vitrification (DV) protocol [46] (Figure 6) with 12 treatment combinations. Due to the limited number of seeds available and lack of previous research for this species, the choice was made to reduce replication in favour of increasing the number of different treatment combinations tested. Seeds were washed by hand-agitation in 1% *w*/*v* Alconox^®^ powdered detergent solution (containing sodium dodecyl benzenesulfonate, tetrasodium diphosphate and sodium carbonate) for 10 min, then sterilized in 2% NaClO solution for 10 min and triple-rinsed in sterile deionized water. Sterilization time was doubled to 10 min for this cryopreservation experiment after poor results in the sterilization test described above.

Seeds were sown on half-strength Murashige and Skoog medium [45] (1/2MS medium, Table 3) with 0.09 M, 0.4 M or 0.8 M sucrose for osmotic desiccation and incubated in darkness at 25 °C for 48 h before cryopreservation. After the desiccation period, seeds were incubated for 20 min in loading solution (1/2MS nutrients, 0.4 M sucrose, 92.1 g/L glycerol, pH 6) at 23 °C, and for 0 or 30 min at 0 °C in plant vitrification solution 2 (PVS2: 0.4 M sucrose, 30% *w*/*v* glycerol, 15% *w*/*v* ethylene glycol, 15% *w*/*v* dimethyl sulfoxide, pH 6 [47]). All cryopreservation solutions were sterilized in an autoclave (Systec DX-150) at 121 °C for 15 min, except for the dimethyl sulfoxide in PVS2 which was filter-sterilized and added after the rest of the solution had cooled. Following incubation in PVS2, seeds were placed in 10 μL droplets of PVS2 on pre-chilled strips of aluminum, placed inside a pre-chilled 2 mL cryo-vial, submerged in liquid nitrogen (+LN) for 1 h, then washed in washing solution (1/2MS nutrients, 1 M sucrose, pH 6) for 20 min at 23 °C. A set of control replicates were placed directly in washing solution after PVS2 treatment without submersion in liquid nitrogen (-LN). Each treatment combination was applied to four or five replicates of five seeds. Following cryopreservation, the seeds were sown on 1/2MS+GA+Z medium, based on a recovery medium previously developed [48] (Table 3).

Seeds were incubated in a growth room at 25 °C in darkness for one month before exposure to fluorescent lighting (Philips Master TL5 HE 35W, cool white) with a 16 h photoperiod (PPFD: 45–50 µmol/m^2^/s). Seeds were monitored weekly for nine weeks for signs of survival (a visual assessment of colour and texture), regeneration (production of a root and/or shoot), and contamination. This monitoring period encompassed an expected germination period of up to six weeks, as published by Floyd [49] and Dunphy et al. [50]. A cut test was used to confirm fill and viability for seeds that had not germinated at the end of the experiment. Germinated seeds were extracted from the Petri dishes and transferred to a commercial potting mix (Annan trial mix AFP > 20%, Grange Growing Solutions, Ebenezer, Australia), which had been steam-pasteurized at 60 °C for a minimum of 30 min. Seedlings were grown in a glasshouse maintained at approximately 26 °C and 85% RH and hand-watered as required.

### 4.3. Initiation of Tissue Cultures

Tissue culture collections of *R. psidioides* were initiated and multiplied using a protocol previously developed at the Australian PlantBank [2] both as an alternative conservation strategy and to produce shoot tips for cryopreservation. Cuttings were taken in late November 2019 from three accessions of three- to six-month-old seedlings in the ABGMA nursery. The seedlings, which had been grown from different seed batches, were healthy and actively growing and had never shown signs of pest or disease infection, including myrtle rust infection. Cuttings were defoliated and reduced to approximately 5 cm lengths with one or two bud pairs each, then separated into two groups: softwood (with apical or axillary buds) and semi-hardwood (with axillary buds only). All cuttings were washed in 2% *w*/*v* Alconox^®^ solution for 30 min while agitated by orbital shaker. Softwood cuttings were then sterilized for 30 min in 1% NaClO solution, and semi-hardwood cuttings in 2% NaClO solution, while agitated by orbital shaker then triple-rinsed in sterile deionised water and trimmed to 1 cm lengths with a single bud pair. The lower concentration NaClO solution was used for softwood cuttings to prevent damage of the more sensitive tissues, based on previous experience with similar cutting material.

A total of 146 explants derived from the cuttings were then sown in individual test tubes (21 × 150 mm) containing 20 mL of 1/2MS medium (Table 3) and placed in the growth room at 25 °C with a 16 h photoperiod (PPFD: 45–50 µmol/m^2^/s). Surviving uncontaminated explants were transferred after 1 month to individual 30 mL polypropylene vials containing 10 mL of MS+Fe+BA+IBA medium (Table 3). After a further two months, surviving explants were subcultured and transferred to 375 mL glass culture jars (Austratech #C936) containing approximately 50 mL of MS+Fe+BA+IBA medium, with the addition of 2 g/L activated charcoal to the medium once every six months. Six to 12 explants (depending on leaf size) from the same accession were combined in each jar. Collections were typically subcultured every 6–8 weeks thereafter.

### 4.4. Cryopreservation of Shoot Tips

A total of 149 apical shoot tips (STs), 1–2 mm long, were excised under sterile conditions from 18-month-old tissue culture collections of *R. psidioides* which had been subcultured eight weeks prior. A single replicate of six STs was sown on 1/2MS medium (Table 3) as a dissection control; the remaining STs were sown on WA medium (Table 3) in preparation for cryopreservation treatments the following day. All STs were then incubated in darkness at 25 °C overnight. On the following day, one third of the STs were sown on 1/2MS medium with 0.4 M sucrose (Table 3) and one third on 1/2MS medium with 0.8 M sucrose for osmotic desiccation (preculture) and incubated in darkness at 25 °C for 48 h. The remaining STs were used for cryopreservation treatments that did not include a desiccation preculture step.

STs were cryopreserved using a DV protocol (Figure 6) with 24 treatment combinations. These consisted of the three desiccation treatments described above in combination with four different incubation times in PVS2 (0, 20, 30 or 60 min) at 0 °C, with half the treatments subsequently immersed in LN. Following cryopreservation, the STs were washed in washing solution for 20 min at 23 °C then sown on 1/2MS medium, with two replicate Petri dishes containing three STs per treatment combination. STs were incubated at 25 °C in darkness for three weeks before exposure to fluorescent light (conditions as described in Section 4.2) and monitored weekly for 22 weeks for signs of survival (presence of green tissue), regeneration (any production of new leaves), and contamination. Experiments were terminated earlier if no signs of survival had been noted for at least three weeks.

### 4.5. Analysis

Differences in the effect of cryopreservation treatments on survival and regeneration of seed and shoot tip material were subjected to analysis of variance and generalized linear modelling using R statistical software (R version 4.1.1) [51].

## Figures and Tables

**Figure 1 plants-14-00699-f001:**
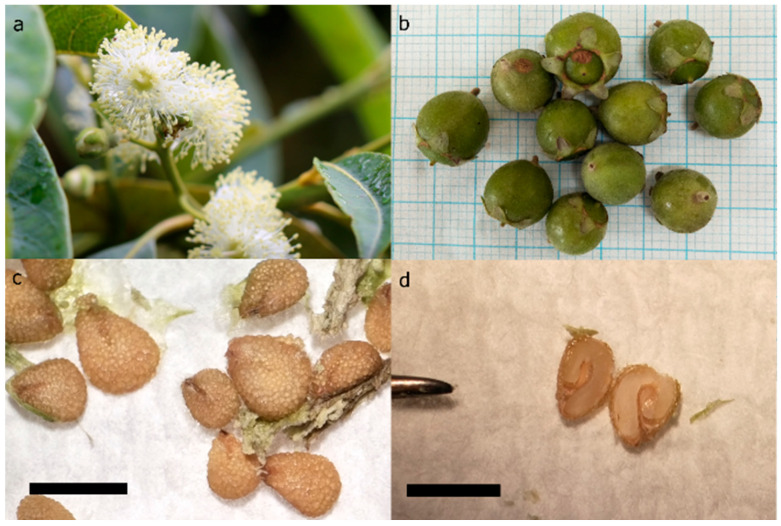
*Rhodomyrtus psidioides* flowers (**a**), fruit (**b**), seeds (**c**), and cross-section of seed showing a healthy embryo inside (**d**). Specimen grown at the Australian Botanic Garden Mount Annan. Bold gridlines in panel (**b**) at 1 cm intervals; scale bar = 2 mm in panels (**c**,**d**).

**Figure 2 plants-14-00699-f002:**
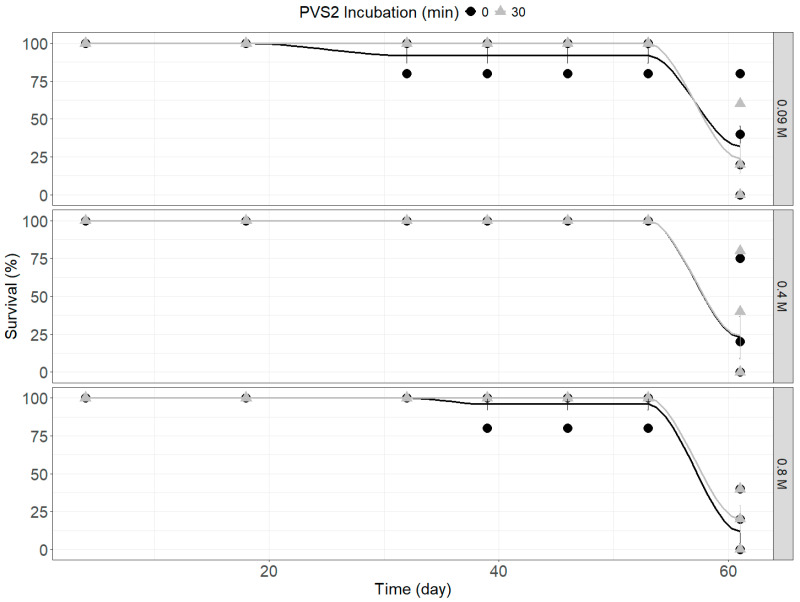
Survival of *Rhodomyrtus psidioides* seeds over time after preculture for 48 h on 1/2MS basal media with 0.09, 0.4, or 0.8 M sucrose and incubation in plant vitrification solution 2 (PVS2) for 0 or 30 min at 0 °C. Points represent individual values for each replicate and fitted curves represent the mean value of three replicates.

**Figure 3 plants-14-00699-f003:**
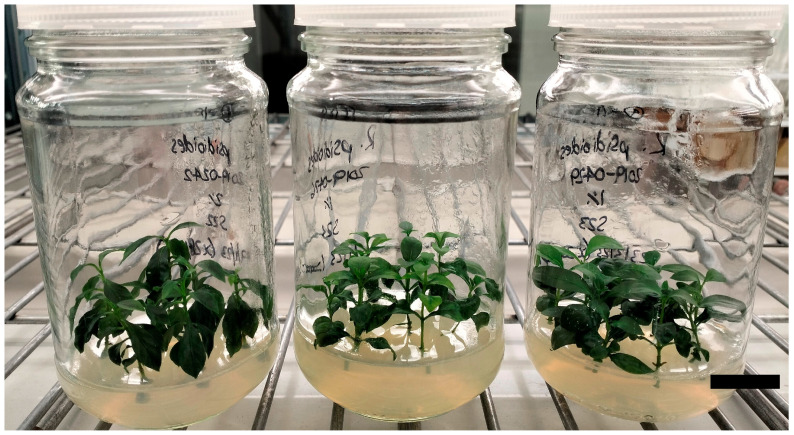
Example jars of *Rhodomyrtus psidioides* tissue culture collections three years after initiation. Explants are growing in full-strength MS medium with double iron concentration and the addition of 2 µM BA and 0.2 µM IBA. Scale bar = 25 mm.

**Figure 4 plants-14-00699-f004:**
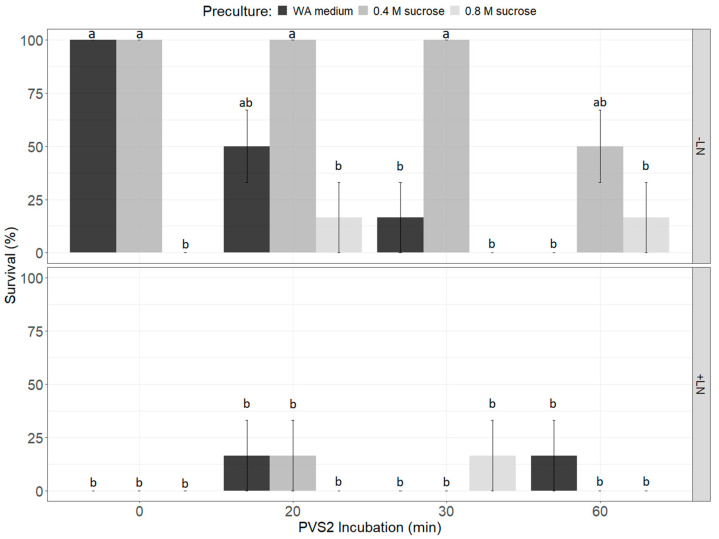
(Top) Survival of *Rhodomyrtus psidioides* shoot tips precultured for 48h on WA medium or 1/2MS medium with 0.4 or 0.8 M sucrose then incubated in PVS2 for 0–60 min at 0°C; and (bottom) survival of shoot tips following preculture, incubation in PVS2 and immersion in liquid nitrogen. Different superscripts above bars represent significant differences among treatments.

**Figure 5 plants-14-00699-f005:**
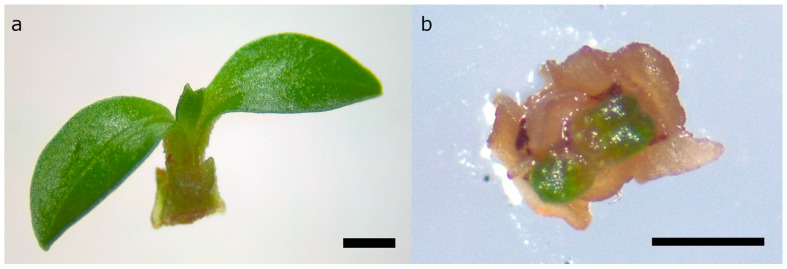
Regenerated shoot tips of *Rhodomyrtus psidioides* with new leaves (**a**) two months after excision; and (**b**) three months after preculture on 0.4 M sucrose basal medium, 20 min incubation in PVS2 at 0 °C, and immersion in LN. Scale bars = 1 mm.

**Figure 6 plants-14-00699-f006:**
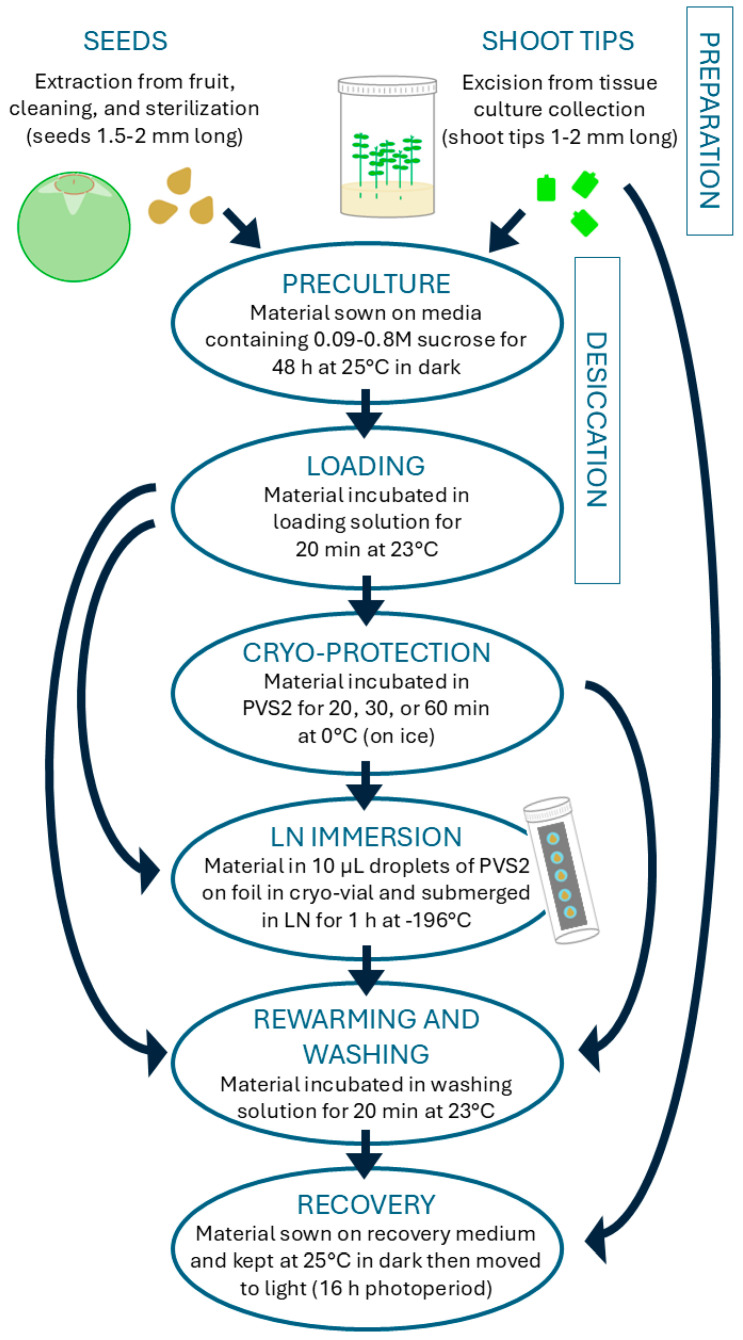
Droplet-vitrification protocol used for cryopreservation of *Rhodomyrtus psidioides* seeds and shoot tips. Arrows show sequence of steps, including the pathway of treated controls.

**Table 1 plants-14-00699-t001:** Mean regeneration (± SD) and surviving seedlings as a percentage of *Rhodomyrtus psidioides* seeds subjected to different cryopreservation treatment combinations. There were five replicates of five seeds (*n* = 25) for each LN- treatment combination and four replicates of five seeds (*n* = 20) for each LN+ treatment combination. Regeneration indicates the production of a root and/or shoot at any time during the experiment (61 days). Surviving seedlings is the percentage of regenerating seeds that grew healthy root systems and true leaves in potting mix after cryopreservation treatments.

Desiccation Treatment	PVS2 Incubation (min)	±LN	Regeneration (%)	Surviving Seedlings (%)
0.09 M sucrose	0	−	16 ± 17	12
0.09 M sucrose	0	+	0	0
0.4 M sucrose	0	−	23 ± 31	16
0.4 M sucrose	0	+	0	0
0.8 M sucrose	0	−	16 ± 17	8
0.8 M sucrose	0	+	0	0
0.09 M sucrose	30	−	24 ± 22	12
0.09 M sucrose	30	+	0	0
0.4 M sucrose	30	−	24 ± 36	8
0.4 M sucrose	30	+	0	0
0.8 M sucrose	30	−	20 ± 20	12
0.8 M sucrose	30	+	0	0

**Table 2 plants-14-00699-t002:** *Rhodomyrtus psidioides* explants initiated into tissue culture and number of surviving explants transferred to multiplication medium after one month. Softwood explants were sterilized in solution containing 1% NaClO and semi-hardwood explants in solution containing 2% NaClO.

Accession No.	No. Softwood Explants (No. Surviving Explants)	No. Semi-Hardwood Explants (No. Surviving Explants)
2019-0242	20 (11)	32 (30)
2019-0426	16 (8)	26 (26)
2019-0429	39 (24)	13 (13)

**Table 3 plants-14-00699-t003:** Basal media used in tests of in vitro growth of *Rhodomyrtus psidioides* seeds at the Australian PlantBank. MS: Murashige and Skoog basal medium [45], BA: 6-benzyladenine, IBA: indole-3-butyric acid, GA3: gibberellic acid. Media were sterilized in an autoclave (Systec DX-150, Wettenberg, Germany) at 121 °C for 15 min.

Medium Name	Medium Description	Plant Growth Regulators
WA	Water agar medium without any added sugar or nutrients, made with 9 g/L agar and pH not adjusted	None
1/2MS	Half-strength MS medium with 30 g/L sucrose, 9 g/L agar, and pH adjusted to 5.75 prior to sterilization	None
MS	Full-strength MS medium with 30 g/L sucrose, 9 g/L agar, and pH adjusted to 5.75 prior to sterilization	None
MS+Fe+BA+IBA	Full-strength MS medium with double iron concentration, made with 30 g/L sucrose, 9 g/L agar, and pH adjusted to 5.75 prior to sterilization	2 µM BA + 0.2 µM IBA
1/2MS+GA+Z	Half-strength MS medium with 20 g/L sucrose, 9 g/L agar, and pH adjusted to 6.00 prior to sterilization	1 µM GA_3_, 0.1 µM Zeatin

## Data Availability

The raw data supporting the conclusions of this article will be made available by the authors upon request.

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
