# Peer review of "Advances Towards Ex Situ Conservation of Critically Endangered Rhodomyrtus psidioides (Myrtaceae)"

_plants, 2025, doi:10.3390/plants14050699_

Round 1
Reviewer 1 Report
Comments and Suggestions for Authors
The paper very realistically demonstrated the issue of contamination in in vitro seed germination and the issue of cryopreservation of an endangered species. It is clearly written and understandable, and I recommend it for publication. Please see just a few suggestions below:
Table 1, which is mentioned earlier in the text, should be before Figure 2 in the results
Line 341: Please write the conditions for tissue culture
Line 144-145: “although the growth of this ST was much slower than that of a control ST (Figure 5)”
I'm curious if you were able to get elongated shoot from the shoot tip that were immersed in LN?
Figure 5b – it is not actually regenerated, but survived shoot tip after treatment with LN (so, improve the Figure description). Have you a photo of single regenerated shoot?
Reviewer 2 Report
Comments and Suggestions for Authors
The paper investigates the long-term conservation of a critically endangered rainforest species from the east coast of Australia, Rhodomyrtus psidioides (G.Don) Benth., through seed and shoot tip cryopreservation. The study is of interest and necessary for conserving these plant genetic resources. Below I have suggested many points that would make the document even more valuable, most of which relate to the need for additional information on materials and methods to make it easier for others to use the proposed procedure and to present the statistical analysis of the data. Another important point relates to shoot tip regrowth X shoot tip survival (I have described this below in the review report). I have also suggested some additional points to be explored in the discussion and the addition of a final sentence to the results section and the addition of a conclusion.
L 106: add the temperature of PVS2 during incubation
Table 1: Add the number of samples which are represented by these data (N) and the statistical analyses.
L 114: Consider using shoots instead of cuttings - please check this throughout the document.
L 118: Make sure you have described this type of bud in the Materials and Methods section.
L 119-123: Add this to the Materials and Methods section – this is missing
Table 2: Add the statistical analyses.
Figure 3 – Add a bar scale and the age of the in vitro plants and the medium in which they are inoculated.
L 132: generalized linear model (GLM)
L 141: add the temperature of PVS2 during incubation
L 144-145: add numbers / %
Figure 4 – Add letters to the treatments to show the statistical difference - there are bars, but letters are more visible to the general public.
Please add the data on shoot tip regrowth!
Figure 5 b is showing survival and not regrowth….
L 232: This means that one shoot tip represents 17%?
L 236: How about embryos? Did the authors try embryos instead of whole seeds? I am still thinking about how whole seeds were cryopreserved by droplet-vitrification.
L 251: Consider adding - Alleviation of oxidative stress and freeze injury by the exogenous addition of various enzymatic and non-enzymatic antioxidants in shoot tip pre-treatment and preculture media – some recent references to support this https://doi.org/10.3390/agronomy13010219 https://doi.org/10.1007/s11240-020-01846-x
L 272: Add a final sentence highlighting the need for further efforts to conserve this species and the main results found in this study.
L 275: Consider adding the period of the year in which the seeds were harvested. Please indicate how long the seed was used after harvesting.
L 292: Add the dimension (size) of the seeds
L 292: Add a figure showing the cryoprocedure – Just think of the way the droplet vitrification was carried out using the whole seed.
L 304-305: Add the pH of the loading solution and PVS2 and how these solutions were sterilized
L 308: Same comment as above for the unloading solution
L 321: Has this pot mix been sterilized?
Is there an effort to cryopreserve embryos instead of whole seeds?
L 331: Cuttings? Or Shoots?
L 340: 1/2MS medium (Table 3)
L 345: medium (Table 3)
L 345: consider adding the number of shoots and the size of jars
L 347: Authors can include this in the figure showing the process - one route for seeds and another for shoot tips.
L 348: Add the age of the in vitro plants from which the shoot tips were isolated.
L 350: 1/2MS medium (Table 3)
L 350: WA medium (Table 3)
L 353: “0.4 M sucrose basal medium and one third on 0.8 M sucrose basal medium” – please describe the medium, liquid or semi-solid (amount of agar), pH etc
L 354: desiccation (preculture)
L 358: add the temperature of PVS2 during incubation
L 359: 1/2MS medium (Table 3)
L 361: fluorescent light (conditions as described in section 4.2)
L 362: Some species may show leaf development in cryopreserved shoot tips without further development - so please describe what regeneration was considered.
L 368: add a conclusion
Round 2
Reviewer 2 Report
Comments and Suggestions for Authors
The authors have done a great job in reviewing the document and all points have been clarified.